# Descent with Misaligned Gradients and Applications to Hidden Convexity

**Aditya Bhaskara**
University of Utah
Salt Lake City, UT
bhaskaraaditya@gmail.com

**Ashok Cutkosky**
Boston University
Boston, MA
ashok@cutkosky.com

**Ravi Kumar**
Google Research
Mountain View, CA
ravi.k53@gmail.com

**Manish Purohit**
Google Research
Mountain View, CA
mpurohit@google.com

## Abstract

We consider the problem of minimizing a convex objective given access to an oracle that produces "misaligned" stochastic gradients, where the expected value of the output is guaranteed to be only correlated, but not necessarily equal to the true gradient of the objective. In the case where the misalignment (or bias) of the oracle changes slowly, we obtain an optimization algorithm that identifies an $\epsilon$-suboptimal point with the optimal iteration complexity of $\tilde{O}(\epsilon^{-2})$; for the more general case where the changes need not be slow, we obtain an algorithm with $\tilde{O}(\epsilon^{-3})$ iteration complexity. As an application of our framework, we consider optimization problems with a "hidden convexity" property, and obtain an algorithm with $O(\epsilon^{-3})$ iteration complexity.

## 1 Introduction

The stochastic gradient descent (SGD) algorithm has been the workhorse of modern machine learning applications due to its efficiency and effectiveness in optimizing complex models. Unlike vanilla gradient descent, SGD operates by iteratively updating model parameters but using only a subset of the data in each iteration. In addition to significantly saving computational cost and enabling it to scale to larger models, the stochastic aspect of SGD helps escape local minima. The convergence of SGD is guaranteed by the fact that the gradient estimate, obtained in each iteration, is *unbiased*.

However, many applications lack easily accessible unbiased or low-variance gradient estimators, particularly when considering quantization effects, approximate preconditioning matrices, or explicit noise addition to achieve differential privacy (Abadi et al., 2016). Unbiased gradient estimators can also be computationally expensive and there is often a trade-off between computational costs, bias, and the variance of the gradient estimators. The topic of biased gradients and understanding the trade offs have been extensively studied in the literature; see, e.g., Hu et al. (2023); Ajalloeian and Stich (2020); Chen and Luss (2018); Sahu et al. (2021); Condat et al. (2022); Hu et al. (2021); Bottou et al. (2018); Beznosikov et al. (2023); Hu et al. (2019; 2020a;b; 2016); Stich and Karimireddy (2020); Karimi et al. (2019); Zhang et al. (2024); Hallak and Levy (2024). See also the recent survey of Demidovich et al. (2023), which unifies many of the existing works, and the references therein.

Several of the existing models for biased or noisy gradients assume an "additive" form of noise, i.e., the gradients are additively perturbed; these additive errors naturally accumulate in the resulting bounds and do not vanish as the number of iterations grows. In this work, we depart from this additive noise model and instead study the setting when the obtained gradient is misaligned because it is weakly correlated with the true gradient. An interesting aspect of our results is that we can obtain convergence guarantees even when the gradients have a significant bias at the optimal point, which is not possible in many previous analyses (Ajalloeian and Stich, 2020; Beznosikov et al., 2023). While our results are closer to some of the parameter settings in Demidovich et al. (2023), to the best of our knowledge, their bounds are either in the general non-convex setting (where the guarantees are weak), or they make much stronger assumptions, such as the PL condition or strong convexity. Our study is also related to the recent body of work on online optimization with hints, where a weakly correlated hint vector was sufficient to obtain logarithmic regret (Dekel et al., 2017; Bhaskara et al., 2020; 2021); however, in those settings, the algorithm sees the unbiased gradient after each step and can thus "correct itself", which is not possible in our setting.

**Contributions and Applications.** We present a high level overview of our results here and defer the formal statements to the respective sections. All of our results consider an objective function $f : \mathbb{R}^d \mapsto \mathbb{R}$ for which we are given a "misaligned" gradient oracle $h$[1] such that $\mathbb{E}[h(x)]$ may not be equal to $\nabla f(x)$, but $\langle \mathbb{E}[h(x)], \nabla f(x) \rangle \geq 0$, for input $x$. That is, $\mathbb{E}[h(x)]$ points in roughly the "same direction" as the true gradient $\nabla f(x)$[2], but it may be misaligned.

Optimization algorithms with misaligned gradients are broadly applicable. Below, we list three applications, each of which requires a different algorithmic treatment, which we present in this paper.

(I) SGD WITH APPROXIMATE PRECONDITIONING: Preconditioning involves transforming the gradient by multiplying with a matrix, and has become of increasing interest in stochastic optimization (see, e.g., Shampoo (Gupta et al., 2018) and variants (Wang et al., 2024; Vyas et al., 2025), AdaHessian (Yao et al., 2021) or Sophia (Liu et al., 2024)). In these works, the gradient is multiplied by a preconditioning matrix, which one hopes is related to a principled preconditioner such as the inverse Hessian or gradient covariance, but in general will only be an estimate. We can formally write $\mathbb{E}[h(x)] = A(x) \cdot \nabla f(x)$ for some SPD preconditioning matrix $A(x)$. Since $A(x)$ is SPD, it is clear that $h(x)$ is correlated with $\nabla f(x)$. Moreover, we might even hope for additional structure in the matrix $A(x)$: since the target ideal preconditioner changes slowly with $x$ in many settings, we might hope that $A(x)$ also changes slowly. In this case, we prove that a variant of projected SGD with momentum converges at the optimal rate of $\tilde{O}(N^{-1/2})$ for any slowly-varying preconditioning scheme (Section 3), where $N$ is the number of gradient evaluations. Notably, typical analyses of stochastic preconditioning methods do not employ momentum despite its use in practice, and require disparate analysis techniques for different preconditioners. Our analysis in contrast critically uses momentum to take advantage of any slowly-varying preconditioning scheme. Note that the $O(N^{-1/2})$ rate is unprovable even for error-free gradient oracles (Nesterov et al., 2018).

(II) COMPRESSION SCHEMES IN DISTRIBUTED OPTIMIZATION: In a distributed setting, gradients are often compressed to reduce expensive communication. Common compression schemes such as top-$k(\cdot)$ lead to biased gradients (Sun et al., 2017). For example, consider a simple compression scheme that extracts and returns the highest-magnitude entry of the gradient. Clearly, this scheme is not unbiased, but it will be correlated. This application is similar to a setup previously studied by Beznosikov et al. (2023, Definition 2); our assumptions in Section 4 are essentially identical, but our setting does not require the noise to go to zero when the true gradient goes to zero. Our work also improves previous analyses (e.g., (Beznosikov et al., 2023; Demidovich et al., 2023)) either by relaxing strong convexity requirements or by providing better convergence guarantees.

(III) HIDDEN CONVEXITY: Suppose $f : \mathbb{R}^d \to \mathbb{R}$ is a function that is non-convex, but there exists an invertible, potentially non-linear, coordinate transformation function $P : \mathbb{R}^d \to \mathbb{R}^d$ such that $f(x) = C(P(x))$ for a convex function $C : \mathbb{R}^d \to \mathbb{R}$. This notion of *hidden convexity* was recently studied by Fatkhullin et al. (2023) and captures many applications in different areas such as reinforcement learning (Sun and Fazel, 2021), revenue management (Chen et al., 2024), and training neural networks (Ergen and Pilanci, 2021; Wang et al., 2022; Sakos et al., 2024), among others. In this case, the minimizer of $f$ is also the minimizer of $C$, but we do not have direct access to $C$, so we cannot perform gradient descent. However, $\nabla f(x)$ can be seen to be $J(x)^T \nabla C(P(x))$, where $J(x)$ is the Jacobian of the transformation $P$ at $x$. Thus, having access to a stochastic oracle for $\nabla f$ is equivalent to having one for $C$, where the gradient is transformed by a Jacobian matrix in expectation. The challenge in this case is that we cannot evaluate $C$ at arbitrary points and instead must access it only through $C(P(x))$ without even knowing the value of $P(x)$. In Section 5, we develop a new algorithm and analysis, yielding a convergence rate of $N^{-1/3}$. Prior work either assumes different smoothness conditions, access to the raw Jacobian $J(x)$, or achieves worse convergence rates.

**Organization.** In Section 3, we assume that the expected gradient is obtained by multiplying the true gradient with an (unobserved) SPD matrix; further, these matrices do not change much over time. In Section 4, we consider a more general setup, where the expected gradient is simply assumed to be correlated with the true gradient. In Section 5, we consider the setting of hidden convexity, where we wish to minimize a (non-convex) function $f$ of the form $f(x) = C(P(x))$, where $C$ is convex and

---

[1] Given a point $x$, $h$ outputs a random vector $h(x)$; see Section 3 for details on the setup.

[2] when $f$ is convex, we let $\nabla f(x)$ indicate an arbitrary subgradient at $x$

$P$ is a non-linear coordinate transform. Here, we obtain an unbiased estimate of the gradient of $f$; however, due to the transformation, this can be viewed as a misaligned gradient for $C$.

## 2    NOTATION AND PRELIMINARIES

Throughout, we deal with functions $f : \mathbb{R}^d \to \mathbb{R}$, where $d$ is the dimension, and we denote by $x_\star$ the minimizer, i.e., $x_\star \in \text{argmin} f$. We assume $\|x_\star\| \leq R$ for some scalar $R$. We write $[T] = \{1, \ldots, T\}$ for any integer $T \geq 1$. Unless otherwise specified, $\|x\| = \|x\|_2$ for any vector $x \in \mathbb{R}^d$ denotes the usual Euclidean norm. For any symmetric positive definite (SPD) matrix $M$, we define the *matrix-induced norm* as $\|v\|_M^2 = v^\top M v$. The *operator norm* of a matrix $M$ is defined as $\|M\|_{\text{op}} = \inf_c\{\|Mx\| \leq c\|x\|, \forall x\}$; when clear from context, we also use $\|M\|$ to denote the operator norm of $M$.

*Projections.* For $D \geq 0$ and $x \in \mathbb{R}^d$, let $\Pi_D[x] = \text{argmin}_{y:\|y\|\leq D} \|x - y\|$ denote the standard $\ell_2$-projection of $x$ onto a Euclidean ball of radius $D$. We also perform projections using the matrix norm: for an SPD $M$, $D \geq 0$, and $x \in \mathbb{R}^d$, let $\Pi_D^M[x] = \text{argmin}_{y:\|y\|_M \leq D} \|x - y\|_M$.

*Function properties.* We say that a function $f$ is $H$-*Lipschitz* if for all $x, y$, we have $|f(x) - f(y)| \leq H \cdot \|x - y\|$; note that this is equivalent to the condition that $\|\nabla f(x)\| \leq H$ for all $x$. We also use the following standard definition of smoothness of a function: $f$ is said to be $L$-*smooth* for some parameter $L$ if for all $x, y$, we have $f(y) \leq f(x) + \langle \nabla f(x), y - x \rangle + \frac{L}{2} \|y - x\|^2$.

*Parameter tuning.* All of our algorithms assume knowledge of various problem parameters (e.g. $H$, $L$, or $R$) that are used to specify optimized learning rates and other hyperparameters. In practice, the learning would need to be adjusted using standard hyperparameter tuning techniques.

## 3    MATRIX-TRANSFORMED GRADIENTS

In this section, we assume that the misaligned stochastic gradients are obtained by a smooth matrix-based transformation. We start by formally stating the assumptions necessary in this section.

**(A1) Oracle & Assumptions.**    Let $f$ be $H$-Lipschitz and convex. We assume that the algorithm has access to an oracle that generates stochastic gradients as follows: given $x$, the oracle returns a random vector $h(x)$ with $\|h(x)\| \leq H$ with probability 1 and $\mathbb{E}[h(x)] = A(x) \cdot \nabla f(x)$ where $A(x) \in \mathbb{R}^{d \times d}$ denotes an SPD matrix that perturbs the stochastic gradients. As outlined earlier, such a transformation ensures the condition $\langle \mathbb{E}[h(x)], \nabla f(x) \rangle \geq 0$. We assume that $\|A(x)^{-1} - A(y)^{-1}\|_{\text{op}} \leq \rho\|x - y\|_2$ for any $x, y \in \mathbb{R}^d$ and assume that the eigenvalues of $A(x)$ lie in $[\lambda_{\min}, \lambda_{\max}]$ for all $x \in \mathbb{R}^d$.

We design an algorithm that, after $N$ queries to the misaligned gradient oracle, outputs a point $x$ such that $f(x) - f(x_\star) \leq \tilde{O}(N^{-1/2})$. Our algorithm can be viewed as projected gradient descent using the misaligned gradients, along with iterate-averaging; see Algorithm 1 for details.

---

**Algorithm 1** Non-Smooth Optimization with Matrix-Transformed Gradients.

---

**Require:** Time horizon $N$, projection radius $D$, learning rate $\eta$.
  1: **Initialize:**
        $x_1 \leftarrow 0, z_1 \leftarrow 0$
  2: **for** $t = 1$ to $N$ **do**
  3:      Generate misaligned gradient estimate: $h_t \leftarrow h(x_t)$
  4:      $z_{t+1} \leftarrow \Pi_D[z_t - \eta h_t]$
  5:      # variables $\hat{g}_t, m_t$ are needed for connection with momentum (not used in analysis)
  6:      $\hat{g}_t \leftarrow (z_t - z_{t+1})/2\eta$
  7:      $m_t \leftarrow \beta_t m_{t-1} + (1 - \beta_t)\hat{g}_t$ with $\beta_t = 1 - 2/(t + 1)$
  8:      $x_{t+1} \leftarrow \frac{1}{t+1} \sum_{i=1}^{t+1} z_i = x_t - \eta m_t$

---

**Connection to Momentum.**    In Algorithm 1, we provide two equivalent forms of our update; the equivalence is demonstrated in Appendix A.1 for completeness. The intermediate variables in lines

6–7 are not used in our analysis, but they provide valuable context for our method. If we were to ignore the projection step in line 4 (which in practice one expects to only rarely be active) then $\hat{g}_t = h_t/2$ and the definition of $m_t$ in line 7 and the second form of the update in line 8 is exactly the same form as standard SGD with momentum. Thus, our analysis in fact shows that the addition of momentum helps correct for gradient misalignment. This is significant because standard theoretical analyses of preconditioned algorithms like Shampoo frequently do not employ momentum, even though the methods used in practice do; our results are a step towards filling this gap.

**Overview of Techniques.** Our analysis is based on several known ideas in stochastic convex optimization such as iterate-averaging and projections, but we need to apply these in a carefully crafted manner to obtain our result. First, our iterate-averaging scheme is not the vanilla Polyak averaging—we query gradients at the running average rather than only averaging once at the end. This gives our iterates a natural "stability" property: $\|x_{t+1} - x_t\| = O(1/t)$, which is critical in our analysis. Secondly, the projection steps are not used to ensure feasibility (as is often the case in constrained optimization), but indeed to control a technical issue in which the iterates move too far from the optimum point. Although not recognized by many analyses, this last challenge is usually not present with unbiased gradient oracles—they never cause the SGD iterates to stray far from the optimal point (Carmon and Hinder, 2022).

Finally, since our algorithm updates the iterates using misaligned stochastic gradients such that $\mathbb{E}[h_t] = A_t g_t$ where $A_t = A(x_t)$ and $g_t = \nabla f(x_t)$, it is convenient for the analysis to work in the matrix-induced norm $\| \cdot \|_{A_t^{-1}}$. Unfortunately, the algorithm does not know the matrix $A_t$, so we cannot perform the projection step in Algorithm 1 using the matrix-induced norm. The key idea is to show that by appropriately setting the projection radius $D$, the $\ell_2$-projection performed in Algorithm 1 is equivalent to projecting via the matrix-induced norm into a *different* domain that nevertheless still contains the optimum $x_\star$.

**Lemma 3.1.** *Suppose $A \in \mathbb{R}^{d \times d}$ is an SPD matrix with all eigenvalues in $[\lambda_{\min}, \lambda_{\max}]$. Let $R > 0$ and let $D = R\sqrt{\frac{\lambda_{\max}}{\lambda_{\min}}}$. Let $x \in \mathbb{R}^d$ be an arbitrary vector. Then there exists a $K$ such that $\Pi_D[x] = \Pi_K^A[x]$ and also for all $x_\star$ with $\|x_\star\|_2 \leq R$, it holds that $\|x_\star\|_A \leq K$.*

The proof relies on showing that our definition of $\Pi_K^A(x)$ is always some scalar multiple of $x$. The condition number guarantees then imply the lemma.

**Theorem 3.2.** *Assume (A1). Let $x_1, \ldots, x_N$ be the iterates produced by Algorithm 1 when we set the projection radius $D = R\sqrt{\frac{\lambda_{\max}}{\lambda_{\min}}}$ and step size $\eta = \frac{\sqrt{R^2/2\lambda_{\min} + 4\rho \mathcal{H}_N (R\sqrt{\lambda_{\max}/\lambda_{\min}})^3}}{H\sqrt{\lambda_{\max} + 1/\lambda_{\min}}\sqrt{N}}$ where $\mathcal{H}_N = \Theta(\log N)$ is the $N$th harmonic number. Then the final iterate $x_N$ satisfies:*

$$\mathbb{E}[f(x_N) - f(x_\star)] \leq 2RH\sqrt{\frac{\left(\frac{1}{2\lambda_{\min}} + 4\rho R \left(\frac{\lambda_{\max}}{\lambda_{\min}}\right)^{3/2} \mathcal{H}_N\right)\left(\lambda_{\max} + \frac{1}{\lambda_{\min}}\right)}{N}} = O\left(\sqrt{\frac{\log N}{N}}\right).$$

*Proof.* We first note that since we use iterate-averaging, the iterates $x_t$ are stable. We have,

$$\|x_t - x_{t-1}\| = \left\|\frac{\sum_{i=1}^{t-1} z_i}{t-1} - \frac{\sum_{i=1}^{t} z_i}{t}\right\| = \left\|\sum_{i=1}^{t-1} \frac{z_i}{t(t-1)} - \frac{z_t}{t}\right\| = \frac{1}{t} \cdot \|x_{t-1} - z_t\| \leq \frac{2D}{t}, \quad (1)$$

where the inequality follows since both $x_{t-1}$ and $z_t$ have norm at most $D$.

Let $g_t = \nabla f(x_t)$ (unknown to the algorithm). Then using anytime online-to-batch conversion (Cutkosky, 2019, Theorem 1), we have the following[3]:

$$\mathbb{E}[f(x_N) - f(x_\star)] \leq \frac{1}{N} \cdot \mathbb{E}\left[\sum_{t=1}^{N} \langle g_t, z_t - x_\star \rangle\right]. \quad (2)$$

To bound the RHS above, we start by analyzing the distance between the $z_t$'s and $x_\star$. By Lemma 3.1, there exists some $K_t \in \mathbb{R}^+$ such that $z_{t+1} = \Pi_{\|z\| \leq D}[z_t - \eta h_t] = \Pi_{K_t}^{A_t^{-1}}[z_t - \eta h_t]$, and

---

[3]For completeness, we include a self-contained proof in Appendix A.3.

$\|x_\star\|_{A_t^{-1}} \le K_t$. (Notice that we are working with the norm induced by $A_t^{-1}$.) Since we assumed that $A_t$ is an SPD matrix with eigenvalues in $[\lambda_{\min}, \lambda_{\max}]$, $A_t^{-1}$ is also symmetric positive and has eigenvalues in $[\lambda_{\max}^{-1}, \lambda_{\min}^{-1}]$. Hence, we have:

$$\|z_{t+1} - x_\star\|_{A_t^{-1}}^2 \le \|z_t - \eta h_t - x_\star\|_{A_t^{-1}}^2$$

$$= \|z_t - x_\star\|_{A_t^{-1}}^2 - 2\eta\langle h_t, A_t^{-1}(z_t - x_\star)\rangle + \eta^2\|h_t\|_{A_t^{-1}}^2. \tag{3}$$

Let $r_t = h_t - \mathbb{E}[h_t] = h_t - A_t g_t$. Then we have $\mathbb{E}[r_t] = 0$ and $\mathbb{E}[\|r_t\|^2] \le \mathbb{E}[\|h_t\|^2] \le H^2$. By substituting $h_t = A_t g_t + r_t$, we can simplify the two terms in (3) as:

$$\|h_t\|_{A_t^{-1}}^2 = (A_t g_t + r_t)^\top A_t^{-1}(A_t g_t + r_t) = g_t^\top A_t g_t + r_t^\top A_t^{-1} r_t + 2\langle g_t, r_t\rangle$$

$$= \|g_t\|_{A_t}^2 + \|r_t\|_{A_t^{-1}}^2 + 2\langle g_t, r_t\rangle,$$

$$\langle h_t, A_t^{-1}(z_t - x_\star)\rangle = \langle A_t g_t + r_t, A_t^{-1}(z_t - x_\star)\rangle = \langle g_t, z_t - x_\star\rangle + \langle r_t, A_t^{-1}(z_t - x_\star)\rangle. \tag{4}$$

Substituting back in (3) and taking expectations (and using $\mathbb{E}[r_t] = 0$) yields:

$$\mathbb{E}[\|z_{t+1} - x_\star\|_{A_t^{-1}}^2] \le \mathbb{E}[\|z_t - x_\star\|_{A_t^{-1}}^2] - 2\eta\langle g_t, z_t - x_\star\rangle + \eta^2\|g_t\|_{A_t}^2 + \eta^2\|r_t\|_{A_t^{-1}}^2. \tag{5}$$

Rearranging and summing:

$$\mathbb{E}\left[\sum_{t=1}^N \langle g_t, z_t - x_\star\rangle\right] \le \mathbb{E}\left[\frac{\|z_1 - x_\star\|_{A_1^{-1}}^2}{2\eta} - \frac{\|z_{N+1} - x_\star\|_{A_N^{-1}}^2}{2\eta}\right.$$

$$\left. + \sum_{t=2}^N \frac{\|z_t - x_\star\|_{A_t^{-1}}^2 - \|z_t - x_\star\|_{A_{t-1}^{-1}}^2}{2\eta} + \frac{\eta}{2}\sum_{t=1}^N\left(\|g_t\|_{A_t}^2 + \|r_t\|_{A_t^{-1}}^2\right)\right]. \tag{6}$$

To simplify, let us consider the following expression:

$$\|z_t - x_\star\|_{A_t^{-1}}^2 - \|z_t - x_\star\|_{A_{t-1}^{-1}}^2 = \|z_t - x_\star\|_{A_t^{-1} - A_{t-1}^{-1}}^2 = \langle z_t - x_\star, (A_t^{-1} - A_{t-1}^{-1})(z_t - x_\star)\rangle$$

$$\le \|z_t - x_\star\|_2 \cdot \|A_t^{-1} - A_{t-1}^{-1}\|_{\mathrm{op}} \cdot \|z_t - x_\star\|_2 = \|z_t - x_\star\|_2^2 \cdot \|A_t^{-1} - A_{t-1}^{-1}\|_{\mathrm{op}}$$

$$\le \rho \cdot \|z_t - x_\star\|_2^2 \cdot \|x_t - x_{t-1}\| \le \rho \cdot (2D)^2 \cdot \frac{2D}{t} = \frac{8\rho D^3}{t},$$

where the three inequalities follow from Cauchy–Schwarz, the assumed bound on the operator norm, and (1) respectively. We can then substitute back into (6) and replace $\sum_{t=2}^N \frac{1}{t} \le \mathcal{H}_N$ where $\mathcal{H}_N = \Theta(\log N)$ is the $N$th harmonic number. We also note that $\|g_t\|_{A_t}^2 \le \|g_t\|_2^2\|A_t\|_{\mathrm{op}}$ and $\|r_t\|_{A_t^{-1}}^2 \le \|r_t\|_2^2\|A_t^{-1}\|_{\mathrm{op}}$. Resuming,

$$\mathbb{E}\left[\sum_{t=1}^N \langle g_t, z_t - x_\star\rangle\right] \le \mathbb{E}\left[\frac{\|z_1 - x_\star\|_{A_1^{-1}}^2}{2\eta} + \frac{8\rho D^3\mathcal{H}_N}{2\eta} + \frac{\eta}{2}\sum_{t=1}^N\left(\|g_t\|_{A_t}^2 + \|r_t\|_{A_t^{-1}}^2\right)\right]$$

$$= \frac{\|z_1 - x_\star\|_{A_1^{-1}}^2}{2\eta} + \frac{8\rho D^3\mathcal{H}_N}{2\eta} + \frac{\eta}{2}\sum_{t=1}^N \mathbb{E}\left[\|g_t\|_{A_t}^2 + \|r_t\|_{A_t^{-1}}^2\right]$$

$$\le \frac{\|z_1 - x_\star\|_{A_1^{-1}}^2}{2\eta} + \frac{8\rho D^3\mathcal{H}_N}{2\eta} + \eta N H^2(\max_t\|A_t\|_{\mathrm{op}} + \|A_t^{-1}\|_{\mathrm{op}})$$

$$\le \frac{R^2}{2\eta\lambda_{\min}} + \frac{8\rho\lambda_{\max}^{3/2}R^3\mathcal{H}_N}{2\lambda_{\min}^{3/2}\eta} + \eta N H^2\left(\lambda_{\max} + \frac{1}{\lambda_{\min}}\right).$$

To balance out the terms, we set the learning rate $\eta$ as:

$$\eta = \frac{\sqrt{R^2/2\lambda_{\min} + 4\rho\mathcal{H}_N(R\sqrt{\lambda_{\max}/\lambda_{\min}})^3}}{H\sqrt{\lambda_{\max} + 1/\lambda_{\min}}\sqrt{N}},$$

and get the following, which along with (2) completes the proof:

$$\mathbb{E}\left[\sum_{t=1}^N \langle g_t, z_t - x_\star\rangle\right] \le 2RH\sqrt{N\left(\frac{1}{2\lambda_{\min}} + 4\rho R\left(\frac{\lambda_{\max}}{\lambda_{\min}}\right)^{3/2}\mathcal{H}_N\right)\left(\lambda_{\max} + \frac{1}{\lambda_{\min}}\right)}. \quad \square$$

## 4 GENERAL MISALIGNED GRADIENTS

In the previous section, we assume that at time $t$, the gradient oracle returns a vector $h_t$ such that $\mathbb{E}[h_t] = A_t g_t$ and further assume that the matrices $A_t = A(x_t)$ change slowly over time. We now consider a more general settin in which $\mathbb{E}[h_t]$ is merely positively correlated with the gradient $g_t$, which is a weaker assumption. In particular, this allows $\mathbb{E}[h_t] \neq 0$ even when $g_t = 0$. We formally specify the gradient oracle and our assumptions below.

**(A2) Oracle & Assumptions.** Let $f : \mathbb{R}^d \to \mathbb{R}$ be a $H$-Lipschitz convex function. Let $x_\star \in$ argmin $f$ satisfy $\|x_\star\| \leq R$ for some known parameter $R$. Let $h$ be the misaligned stochastic gradient oracle for $f$. We assume that the oracle $h$ satisfies the following *correlation conditions*: for any input $x$, $h(x)$ outputs a random vector such that:

- $\|h(x)\| \leq H$ with probability 1,
- The expected output $\xi(x) := \mathbb{E}[h(x)]$ satisfies the condition $\langle \nabla f(x), \xi(x) \rangle \geq \alpha \|\nabla f(x)\| \|\xi(x)\|$ for some parameter $\alpha$, and
- $\|\xi(x)\| \geq \beta \|\nabla f(x)\|$, for some (possibly unknown) parameter $\beta$.

We assume that outputs for different calls to $h(x)$ on the same input $x$ are independent. It turns out that our arguments hold more generally, but we would need the assumptions above to hold conditioned on the "query history". To simplify the presentation, we will assume independent outputs.

We now present our result for optimizing smooth convex functions, given access to a stochastic oracle that outputs a misaligned gradient, subject to the correlation conditions defined above. In addition to the assumptions above, we assume that $f$ is $L$-smooth (see Section 2).

The main result—stated formally in Theorem 4.4—shows that after $N$ queries to the misaligned gradient oracle, the algorithm obtains a point $x$ such that $f(x) - f(x_*) \leq O(N^{-1/3})$. Our algorithm can be viewed as SGD with mini-batches, but with two key modifications: (i) we move along the normalized gradient estimate using a carefully designed step size schedule, and (ii) we add a novel correction step that prevents the iterates from being too far from the optimum. Algorithm 2 presents the full details. In the following, let $[z]_-$ denote $\min(z, 0)$.

Before the analysis, let us briefly compare this result to those in Demidovich et al. (2023), which considered a broadly similar setup. Their results for the convex case either require $\mu$-strong convexity and achieve a convergence rate of $O(1/(\mu^2 N))$, or achieve $O(1/N^{1/4})$ without strong convexity. Our result does not require strong convexity and achieves $\tilde{O}(1/N^{1/3})$. Note that, unlike in standard unbiased optimization, we cannot simply regularize a non-strongly convex objective to become strongly convex because this can destroy the correlation structure in the gradient estimates.

---

**Algorithm 2** Smooth Convex Optimization with Misaligned Gradients.

---

**Require:** Time horizon $T$, norm bound $D$, sequence $B_1, \ldots, B_T$ of minibatch sizes, sequence $\eta_1, \ldots, \eta_T$ of learning rates.
1: **Initialize:**
$\quad x_1 \leftarrow 0$
2: **for** $t = 1$ to $T$ **do**
3: $\quad$ Generate a misaligned gradient estimate using a minibatch of size $B_t$ by calling $h$ on the same point $B_t$ times and taking the average, i.e., $h_t \leftarrow \frac{1}{B_t} \sum_{\tau=1}^{B_t} h(x_t)$
4: $\quad \overline{h}_t \leftarrow \frac{h_t}{\|h_t\|}$ and $\hat{x}_{t+1} \leftarrow x_t - \eta_t \overline{h}_t$
5: $\quad$ **if** $\|\hat{x}_{t+1}\| > D$ **then**
6: $\quad\quad x_{t+1} \leftarrow x_t - \eta_t \overline{h}_t + \eta_t \frac{[\langle \overline{h}_t, x_t \rangle]_- \cdot x_t}{\|x_t\|^2} - \eta_t^2 \frac{x_t}{\|x_t\|^2}$
7: $\quad$ **else**
8: $\quad\quad x_{t+1} \leftarrow \hat{x}_{t+1}$

---

First, we show a few technical statements. The first shows that the norm bound remains bounded throughout. This is a subtle consequence of the extra $-\eta^2 \frac{x_t}{\|x_t\|^2}$ term inn our update for large iterates. Roughly speaking, this extra regularization is "just enough" to keep the norm from growing no matter

what the noisy gradient $h_t$ is. It is possible that a standard projection step would also suffice, but this explicit update is more amenable to our analysis.

**Lemma 4.1.** *Suppose $\eta_t \leq D/2$. Then for all t, $\|x_t\| \leq D$.*

*Proof.* We proceed by induction. Suppose $\|x_t\| \leq D$. Now, if $\|\hat{x}_{t+1}\| \leq D$, then $x_{t+1} = \hat{x}_{t+1}$, so there is nothing to prove. So let us assume that $\|\hat{x}_{t+1}\| > D$. In this case, note that we also have $\|x_t\| \geq \|\hat{x}_{t+1}\| - \eta_t \geq D/2$. Thus, $\eta_t / \|x_t\| < 1$.

Next, define $u_t = \overline{h}_t - \frac{[\langle \overline{h}_t, x_t \rangle]_- \cdot x_t}{\|x_t\|^2}$. By definition, we have $\langle u_t, x_t \rangle \geq 0$ and also $\|u_t\| \leq 1$, since $u_t$ is either $\overline{h}_t$ itself or the projection of $\overline{h}_t$ orthogonal to the direction of $x_t$. Therefore:

$$\|x_{t+1}\|^2 \leq \left( \|x_t\| - \frac{\eta_t^2}{\|x_t\|} \right)^2 + \eta_t^2 \|u_t\|^2 = \|x_t\|^2 - 2\eta_t^2 + \frac{\eta_t^4}{\|x_t\|^2} + \eta_t^2 \|u_t\|^2$$

$$\leq \|x_t\|^2 - \eta_t^2 + \frac{\eta_t^4}{\|x_t\|^2} = \|x_t\|^2 + \eta_t^2 \left( \frac{\eta_t^2}{\|x_t\|^2} - 1 \right) \leq \|x_t\|^2 \leq D. \qquad \square$$

Next, we need the following simple geometric lemma.

**Lemma 4.2.** *Suppose that $g$ and $h$ satisfy the conditions $\langle g, h \rangle \geq \alpha \|g\| \|h\|$ and $\|h\| \geq \beta \|g\|$, for some parameters $\beta > 0$ and $\alpha \in [0, 1]$. Then for any $\delta$, we have*

$$\left\langle g, \frac{h + \delta}{\|h + \delta\|} \right\rangle \geq \frac{2\alpha \|g\|}{3} - \frac{4\|\delta\|}{\beta}. \tag{7}$$

The next lemma gives us the key technical property required for our final analysis. It also shows the dependence on the size of the minibatch, $B_t$.

**Lemma 4.3.** *Let t be any step of the algorithm, and suppose that $\eta_t \leq D/2$ and $D \geq 12R/\alpha$. Then $\mathbb{E}[\langle \nabla f(x_t), x_{t+1} - x_t \rangle] \leq -\frac{\eta_t \alpha \|\nabla f(x_t)\|}{3} + \frac{8\eta_t H}{\beta \sqrt{B_t}}$.*

*Proof.* Throughout this proof, we write $g_t = \nabla f(x_t)$. First, define $\delta_t = h_t - \mathbb{E}[h_t]$, and for convenience, write $\xi_t = \mathbb{E}[h_t]$. Recall the correlation conditions satisfied by $\xi_t$. Since we make $B_t$ queries to the oracle using the same $x_t$ and since $\|h_t\| \leq H$, we have $\mathbb{E}[\|\delta_t\|^2] \leq 4H^2/B_t$ and thus $\mathbb{E}[\|\delta_t\|] \leq 2H/\sqrt{B_t}$. We now consider two cases:

*Case 1:* $\|\hat{x}_{t+1}\| \leq D$. Since $x_{t+1} - x_t = -\eta_t \overline{h}_t$, by applying Lemma 4.2 with $\xi_t$ taking the role of $h_t$ in the lemma statement, we obtain

$$\langle g_t, x_{t+1} - x_t \rangle \leq -\frac{2\eta_t \alpha \|g_t\|}{3} + \frac{4\eta_t \|\delta_t\|}{\beta}.$$

*Case 2:* $\|\hat{x}_{t+1}\| > D$. Since $\eta_t < D/2$, we must have $\|x_t\| \geq D/2$. In this case, $x_{t+1} = x_t - \eta_t \overline{h}_t + \eta_t \frac{[\langle \overline{h}_t, x_t \rangle]_- \cdot x_t}{\|x_t\|^2} - \eta_t^2 \frac{x_t}{\|x_t\|^2}$ and thus

$$\langle g_t, x_{t+1} - x_t \rangle = -\eta_t \langle g_t, \overline{h}_t \rangle + \eta_t \frac{[\langle \overline{h}_t, x_t \rangle]_- \cdot \langle g_t, x_t \rangle}{\|x_t\|^2} - \eta_t^2 \frac{\langle g_t, x_t \rangle}{\|x_t\|^2}. \tag{8}$$

In this case, we first note that if $\langle g_t, x_t \rangle \geq 0$, there is nothing to prove, because the last two terms on the RHS above will be non-positive (since $[z]_- \leq 0$ for any $z$).

Thus, let us assume that $\langle g_t, x_t \rangle < 0$. In this case, we note that because $\langle g_t, x_t - x_* \rangle \geq 0$ (which holds by convexity), we have

$$\frac{\langle g_t, x_t \rangle}{\|x_t\|} \geq \frac{\langle g_t, x_* \rangle}{\|x_t\|} \geq -\frac{\|g_t\| \|x_*\|}{\|x_t\|} \geq -\frac{\alpha \|g_t\|}{6}.$$

In the last step, we used the fact that $\|x_t\| \geq D/2 > \frac{6R}{\alpha}$ by choice, and $\|x_*\| \leq R$. Thus, plugging this into (8), we have

$$\langle g_t, x_{t+1} - x_t \rangle \leq -\eta_t \langle g_t, \overline{h}_t \rangle - \frac{\eta_t \alpha}{6} \frac{[\langle \overline{h}_t, x_t \rangle]_- \cdot \|g_t\|}{\|x_t\|} + \eta_t^2 \frac{\alpha \|g_t\|}{6 \|x_t\|}$$

$$\leq -\eta_t \langle g_t, \overline{h}_t \rangle + \frac{\eta_t \alpha}{6} \|g_t\| + \eta_t^2 \frac{\alpha \|g_t\|}{6 \|x_t\|}.$$

In the second step, we used $[z]_- \geq -|z|$, together with Cauchy–Schwarz. Applying Lemma 4.2 as before, along with the fact that $\eta_t \leq D/2 < \|x_t\|$, we have

$$\langle g_t, x_{t+1} - x_t \rangle \leq -\frac{2\eta_t \alpha \|g_t\|}{3} + \frac{4\eta_t \|\delta_t\|}{\beta} + \frac{\eta_t \alpha}{6} \|g_t\| + \eta_t \frac{\alpha \|g_t\|}{6} = -\frac{\eta_t \alpha \|g_t\|}{3} + \frac{4\eta_t \|\delta_t\|}{\beta}.$$

Thus in both the cases, we have

$$\langle g_t, x_{t+1} - x_t \rangle \leq -\frac{\eta_t \alpha \|g_t\|}{3} + \frac{4\eta_t \|\delta_t\|}{\beta} \implies \mathbb{E}[\langle x_{t+1} - x_t, g_t \rangle] \leq -\frac{\eta_t \alpha \|g_t\|}{3} + \frac{8\eta_t H}{\beta \sqrt{B_t}}. \ \square$$

We can now prove the main convergence guarantee.

**Theorem 4.4.** *Assume (A2). Let $D = 12R/\alpha$, and set*

$$k = \frac{12}{\alpha} \ ; \ B_t = (t+1+k)^2 \ ; \ \eta_t = \frac{6D}{\alpha(t+1+k)}.$$

*At the end of $T$ iterations of Algorithm 2, the total number of gradient evaluations is $N = \Theta(T^3)$ and*

$$\mathbb{E}[f(x_T) - f(x_*)] \leq \frac{5200}{N^{1/3}} \cdot \log T \cdot \left( \frac{RH}{\alpha^2 \beta} + \frac{LR^2}{\alpha^4} \right).$$

*Proof.* As before, we write $g_t = \nabla f(x_t)$. We start by bounding $\|x_{t+1} - x_t\|$. Note that since $k \geq \frac{12}{\alpha}$, the property $\eta_t \leq \frac{D}{2}$ is always satisfied.

If $\|\hat{x}_{t+1}\| \leq D$, then we have $\|x_{t+1} - x_t\| = \eta_t$. Otherwise,

$$x_{t+1} - x_t \leq \eta_t \left( \overline{h}_t - \frac{[\langle \overline{h}_t, x_t \rangle]_- \cdot x_t}{\|x_t\|^2} \right) - \eta_t^2 \frac{x_t}{\|x_t\|^2}.$$

As we saw in the proof of Lemma 4.1, the term in the parentheses is the projection of $\overline{h}_t$ orthogonal to $x_t$, and thus it has norm $\leq 1$. Likewise, because $\|x_t\| \geq D/2 > \eta_t$ in this case, the last term also has a norm bounded by $\eta_t$. This implies that $\|x_{t+1} - x_t\| \leq 2\eta_t$.

The analysis is based on analyzing the potential function $\Phi_t = (t+k)(f(x_t) - f(x_\star))$. By definition,

$$\Phi_{t+1} - \Phi_t = (t+1+k)(f(x_{t+1}) - f(x_t)) + (f(x_t) - f(x_\star)).$$

By then using smoothness and applying Lemma 4.3, we have

$$\mathbb{E}[f(x_{t+1}) - f(x_t)] \leq \mathbb{E}\left[ \langle g_t, x_{t+1} - x_t \rangle + \frac{L}{2} \|x_{t+1} - x_t\|^2 \right] \leq -\frac{\eta_t \alpha \|g_t\|}{3} + \frac{8\eta_t H}{\beta \sqrt{B_t}} + 2L\eta_t^2.$$

Also, convexity implies that $f(x_t) - f(x_*) \leq \langle g_t, x_t - x_* \rangle \leq 2D \|g_t\|$. Now from our choice of the parameters, $\eta_t$ will satisfy the condition $2D = \frac{(t+1+k)\eta_t \alpha}{3}$. Plugging these in, we have

$$\mathbb{E}[\Phi_{t+1} - \Phi_t] \leq (t+1+k) \left( \frac{8\eta_t H}{\beta \sqrt{B_t}} + 2L\eta_t^2 \right).$$

Plugging in the value of $B_t, \eta_t$, and $D$, we have

$$\mathbb{E}[\Phi_{t+1} - \Phi_t] \leq (t+1+k) \left( \frac{8\eta_t H}{\beta(t+1+k)} + 2L\eta_t^2 \right) = \frac{48 \cdot DH}{\alpha \beta (t+1+k)} + \frac{72 \cdot LD^2}{\alpha^2(t+1+k)}$$

$$\leq \frac{2592}{(t+1+k)} \cdot \left( \frac{RH}{\alpha^2 \beta} + \frac{LR^2}{\alpha^4} \right).$$

Summing this over $t$, we get

$$\mathbb{E}[\Phi_T] \leq \Phi_0 + 2592 \cdot \log \left( \frac{T+k}{k+1} \right) \cdot \left( \frac{RH}{\alpha^2 \beta} + \frac{LR^2}{\alpha^4} \right).$$

Since we start at the origin, we have $\Phi_0 \leq kR\|\nabla f(0)\| \leq kRH$, so this term can be ignored by slightly increasing the constant in the second term. Thus, by the definition of the potential,

$$\mathbb{E}[f(x_T) - f(x_*)] \leq \frac{2600}{T} \cdot \log T \cdot \left( \frac{RH}{\alpha^2 \beta} + \frac{LR^2}{\alpha^4} \right).$$

Now, observe that the total number of queries to the oracle is

$$N = \sum_t B_t \leq T \cdot B_T = T(T + 1 + k)^2 < 8T^3,$$

assuming $T > k$. Thus, $T \geq N^{1/3}/2$, and plugging this in, we obtain

$$\mathbb{E}[f(x_T) - f(x_*)] \leq \frac{5200}{N^{1/3}} \cdot \log T \cdot \left( \frac{RH}{\alpha^2 \beta} + \frac{LR^2}{\alpha^4} \right). \qquad \square$$

## 5 HIDDEN CONVEXITY

In this section, we consider the optimization of (non-convex) functions that admit a convex reformulation. We start by recalling the notion of *hidden convexity* defined earlier: $f : \mathbb{R}^d \to \mathbb{R}$ is a function that can be expressed as $f(x) = C(P(x)) \; \forall x$, where $C : \mathbb{R}^d \to \mathbb{R}$ is a convex, $H$-Lipschitz function, and $P : \mathbb{R}^d \to \mathbb{R}^d$ is an invertible, possibly non-linear, coordinate transform function. Let $J(x)$ be the Jacobian (total derivative) of $P$ at $x$. Since $P$ is invertible, we also have that $J(x)$ is invertible for all $x$ (indeed, this only requires local invertibility for $P$).

In contrast to some previous work on this problem (Fatkhullin et al., 2023), we do not assume that the convex function $C$ is smooth—instead we will assume that the transformation $P$ is smooth. Note that the composition $f$ will therefore not be smooth in general. This allows us to more accurately model applications with non-smooth losses (e.g., neural networks with a ReLU layer). Previous studies involving similar non-smooth setups generally achieve weaker results. Sakos et al. (2024) develop algorithms assuming that we have access to $J(x)$, which we forbid, and Chen et al. (2024) provide a method that fits exactly our setting to obtain a convergence rate of $O(1/N^{1/4})$, whereas we obtain a convergence rate of $O(1/N^{1/3})$.

**(A3) Oracle & Assumptions.** We assume that there exist constants $\alpha, \beta, \rho > 0$ such that for all $x, y \in \mathbb{R}^d$, we have $\|J(x)\|_{\text{op}} \leq 1/\alpha$ and $\|J^{-1}(x)\|_{\text{op}} \leq 1/\beta$, and $\|J(x) - J(y)\|_{\text{op}} \leq \rho \|x - y\|$. For technical reasons, we also assume knowledge of parameters $D_1, D_2$ such that for all $\|x\| = D_1$ and $\|y\| = D_2$, $f(y) \geq f(x)$. We have access to a stochastic gradient oracle for $f$ that we call $h$. For any input $x$, $h(x)$ outputs a random vector satisfying (i) $\|h(x)\| \leq H$ with probability 1, and (ii) $\mathbb{E}[h(x)] = \nabla f(x)$. As we discussed earlier, $\nabla f(x) = J(x)^T \nabla C(P(x))$, thus the oracle $h$ provides $\nabla C$ at the point $P(x)$, with a transformation matrix applied.

We first show a couple of technical lemmas on hidden convexity. The first lemma upper and lower bounds the distortion of the distance between any two points due to the coordinate transformation.

**Lemma 5.1.** *For all $x, y \in \mathbb{R}^d$, $\beta\|x - y\| \leq \|P(x) - P(y)\| \leq \frac{\|x-y\|}{\alpha}$.*

Next, let $\mathcal{B}(x, \delta) = \{y : \|x - y\| \leq \delta\}$ denote a ball of radius $\delta$ centered at $x$. The following lemma shows that if the function value does not drop significantly anywhere in a small ball of radius $\delta$ around a point $x$, then $x$ is close (in function value) to being a global minima of $f$. (Thus, hidden convexity implies a guarantee very similar to convexity.)

**Lemma 5.2.** *Let $x$ be an arbitrary point and $D > 0$ be such that $\|x\| \leq D$ and $\|x_\star\| \leq D$. Let $\mathcal{B} = \mathcal{B}(x, \delta)$ be a ball of radius $\delta \leq D$ centered at $x$. Suppose that $\sup_{z \in \mathcal{B}}\{f(x) - f(z)\} \leq \epsilon$ for some $\epsilon > 0$. Then,*

$$f(x) - f(x_\star) \leq \epsilon \left( 1 + \frac{3D}{\alpha\beta\delta} \right).$$

**Algorithm Outline.** At each outer iteration (corresponding to $k$), the algorithm performs an optimization in a ball around $x_1^{k,1}$ of an appropriate radius. If the ball contains $x_*$, the algorithm can terminate. If not, we can guarantee (using Lemma 5.2) that $x_1^{k+1,1}$ is closer to $x_*$. Thus with growing $k$, the centers of the balls are guaranteed get closer and closer to $x_*$.

The key technical lemma is the following:

---

**Algorithm 3** Optimization of Hidden Convex Functions.

1: **Initialize:**
   $x_1^{1,1} \leftarrow 0, k \leftarrow 1$
2: $I_1 \leftarrow 2D_2\rho$
3: **for** $k = 1$ to $K$ **do**
4:    $T_k \leftarrow 2^k$
5:    $\delta_k \leftarrow \frac{1}{\rho\sqrt{T_k}}$
6:    $\eta \leftarrow \frac{\delta_k}{H\sqrt{T_k}}$
7:    **if** $k \neq 1$ **then** $I_k \leftarrow \lceil \frac{20D_2}{\alpha\beta\delta_k} \rceil$
8:    **for** $i = 1$ to $I_k$ **do**
9:       $\mathcal{X}^{k,i} \leftarrow \{x : \|x - x_1^{k,i}\| \leq \delta_k\}$
10:      **for** $t = 1$ to $T_k$ **do**
11:         $h_t^k \leftarrow h(x_t^{k,i})$
12:         $x_{t+1}^{k,i} \leftarrow \Pi_{\mathcal{X}^{k,i}}[x_t^{k,i} - \eta h_t^{k,i}]$
13:      Choose $\overline{x^{k,i}}$ uniformly at random from $\{x_1^{k,i}, \ldots, x_{T_k}^{k,i}\}$
14:      **if** $\|\overline{x^{k,i}}\| \geq D_2$ **then**
15:         $x_1^{k,i+1} \leftarrow D_1 \frac{\overline{x^{k,i}}}{\|\overline{x^{k,i}}\|}$
16:      **else**
17:         $x_1^{k,i+1} \leftarrow \overline{x^{k,i}}$
18:    $x_1^{k+1,1} \leftarrow x_1^{k,T_k+1}$
19: Return $\hat{x} \leftarrow x_1^{K+1,1}$

---

**Lemma 5.3.** *Algorithm 3 ensures for all $k \geq 2$:*

$$\mathbb{E}[f(x_1^{k,1}) - f(x_\star)] \leq 20\frac{HD_2}{\alpha\beta\sqrt{2^k}}.$$

The proof is quite involved (see Appendix C.1). With this, the main result is the following.

**Theorem 5.4.** *Assume (A3). Let $N = \sum_k I_k T_k$ be the total number of gradient evaluations by Algorithm 3 and $\hat{x}$ be the final iterate. Then we have*

$$\mathbb{E}[f(\hat{x}) - f(x_\star)] \leq O\left(\frac{D_2^{4/3}H\rho^{1/3}}{(\alpha\beta)^{4/3}N^{1/3}}\right).$$

## 6 CONCLUSION

We have developed algorithms to minimize a convex function given access to misaligned stochastic gradients. Misalignments can be specified using a linear transformation and may or may not vary smoothly with the input point. We obtain different convergent rates for these two cases. We then use these insights to develop an algorithm for minimizing non-convex functions that have a "hidden convexity" property, i.e., there is a coordinate transformation that can make the function convex.

Our work raises natural open questions about the optimality of the results: can the $\epsilon^{-3}$ iteration complexity in two of our results be improved? In the hidden convex case, improving to $\epsilon^{-2}$ is conceptually very interesting, as it would match the convex case. Another "structural" question is if we can obtain one meta-algorithm from which our different algorithmic results can be derived. We currently require different projection steps and additional tricks to deal with non-PSD transformations. Yet another question would be to weaken some of our assumptions (e.g., bounded gradients).

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

## A  MISSING PROOFS IN SECTION 3

### A.1  EQUIVALENCE OF UPDATE FORMS IN ALGORITHM 1

Our analysis of Algorithm 1 makes use of the "anytime online-to-batch" conversion result of Cutkosky (2019). However, this approach (under the alternative name "primal averaging") was shown to be equivalent to SGD with momentum (Defazio, 2020). Here we reproduce the necessary calculations:

**Proposition A.1.** *For any sequence $z_1, \ldots, z_T$ of vectors and any scalar $\eta$, let $m_0 = 0$ and define $\hat{x}_1 = x_1 = z_1$. Then consider the sequences:*

$$\hat{g}_t = \frac{z_t - z_{t+1}}{2\eta}, \quad m_t = \left(1 - \frac{2}{t+1}\right) m_{t-1} + \frac{2\hat{g}_t}{t+1}, \quad \hat{x}_{t+1} = \hat{x}_t - \eta m_t, \quad x_{t+1} = \frac{\sum_{i=1}^{t+1} z_i}{t+1}.$$

*Then $\hat{x}_t = x_t$ for all $t$.*

*Proof.* First, note that

$$tx_t = (t-1)x_{t-1} + z_t \implies x_t - z_t = (t-1)(x_{t-1} - x_t).$$

We proceed by induction. From direct calculation, $m_1 = (z_1 - z_2)/\eta$, so that $\hat{x}_2 = x_1 + \frac{z_2 - z_1}{2} = \frac{z_1 + z_2}{2} = x_2$. Next, let $x_i = \hat{x}_i$ for some all $i \le t$. Then clearly $\eta m_i = x_{i+1} - x_i$ for all $i < t$. Now,

$$\begin{aligned}
x_{t+1} &= x_t + \frac{z_{t+1} - x_t}{t+1} \\
&= \hat{x}_t + \frac{(z_{t+1} - z_t) + (z_t - x_t)}{t+1} \\
&= \hat{x}_t - \frac{2\eta \hat{g}_t}{t+1} + \frac{x_t - z_t}{t+1} \\
&= \hat{x}_t - \frac{2\eta \hat{g}_t}{t+1} - \frac{(t-1)(x_t - x_{t-1})}{t+1} \\
&= \hat{x}_t - \frac{2\eta \hat{g}_t}{t+1} - \frac{\eta(t-1)m_{t-1}}{t+1} \\
&= \hat{x}_t - \left(1 - \frac{2}{t+1}\right)\eta m_{t-1} - \frac{2\eta \hat{g}_t}{t+1} \\
&= \hat{x}_t - \eta m_t \\
&= \hat{x}_{t+1}. \qquad\qquad\qquad \square
\end{aligned}$$

## A.2 PROOF OF LEMMA 3.1

*Proof.* First, consider the case $\|x\|_2 \le D$. In this case, $\Pi_D[x] = x$, and so we can set $K = \infty$ to achieve $\Pi_K^A[x] = x = \Pi_D[x]$. Furthermore, $\|x_\star\|_A \le K$ for all $x_\star$.

Next, consider the case $\|x\|_2 > D$. We have $\Pi_D[x] = Dx/\|x\|$.

Let $v$ be the projection, so that $v = \operatorname{argmin}_{\|v\|_A \le K} \|v - x\|_A^2$. Therefore by Lagrange multipliers:

$$2A(v - x) = \lambda Av.$$

Therefore $v$ is the point that satisfies for some $\alpha$:

$$\|v\|_A = \sqrt{v^\top A v} = K \quad \text{and} \quad \alpha Av = A(x - v).$$

For $K = D\|x\|_A/\|x\|_2$ and $\alpha = 1 - D/\|x\|_2 \ge 0$ and we can verify that:

$$\|\Pi_D[x]\|_A = \frac{D}{\|x\|_2}\|x\|_A = K,$$
$$A(x - \Pi_D[x]) = A(x - Dx/\|x\|)_2 = A(1 - D/\|x\|_2)x = \alpha Ax.$$

So, this shows that $\Pi_D[x] = \Pi_K^A[x]$. It remains to show that $\|x_\star\|_A \le K$ for all $x_\star$ such that $\|x_\star\|_2 \le R$. To this end, we compute:

$$\|x_\star\|_A \le |x_\star\|_2 \cdot \sqrt{\sup_w \frac{w^\top A w}{\|w\|_2^2}} \le \sqrt{\lambda_{\max}} \cdot \|x_\star\|_2 \le \sqrt{\lambda_{\max}} \cdot R$$

$$= D\sqrt{\lambda_{\min}} \le D \cdot \sqrt{\inf_w \frac{w^\top A w}{\|w\|_2^2}} \le D \cdot \frac{\|x\|_A}{\|x\|_2} = K,$$

where we used the fact that $A$ is symmetric. $\qquad\qquad \square$

## A.3 GRADIENT DESCENT WITH ITERATE-AVERAGING

For the sake of completeness, we now give a short proof of Theorem 1 from Cutkosky (2019) on iterate-averaged online gradient descent that was used in Section 3

**Theorem A.2.** *Suppose $x_t = \frac{1}{t} \sum_{i=1}^{t} z_i$. Then for any convex $F$ and point $u$, for all $T$:*

$$F(x_T) - F(u) \leq \frac{\sum_{t=1}^{T} \langle \nabla F(x_t), z_t - u \rangle}{T}.$$

*Proof.* Define $g_t = \nabla F(x_t)$. Then:

$$\sum_{t=1}^{T} F(x_t) - F(u) \leq \sum_{t=1}^{T} \langle g_t, x_t - u \rangle = \sum_{t=1}^{T} \langle g_t, x_t - z_t \rangle + \langle g_t, z_t - u \rangle$$

Observe that $x_t - z_t = (t-1)(x_{t-1} - x_t)$:

$$= \sum_{t=1}^{T} (t-1) \langle g_t, x_{t-1} - x_t \rangle + \langle g_t, z_t - u \rangle$$

$$\leq \sum_{t=1}^{T} (t-1)(F(x_{t-1}) - F(x_t)) + \sum_{t=1}^{T} \langle g_t, z_t - u \rangle.$$

The last step also follows by convexity. Rearranging and telescoping completes the proof. □

## B   MISSING PROOF IN SECTION 4

*Proof of Lemma 4.2.* We consider two cases:

*Case 1: $\|\delta\| \geq \frac{\|h\|}{2}$.* Here, $\frac{4\|\delta\|}{\beta} \geq \frac{2\|h\|}{\beta} \geq 2\|g\|$. Thus the RHS of (7) is $< -\|g\|$ and the inequality trivially holds by Cauchy–Schwarz.

*Case 2: $\|\delta\| < \frac{\|h\|}{2}$.* Here, we have

$$\frac{3\|h\|}{2} \geq \|h + \delta\| \geq \frac{\|h\|}{2} \geq \frac{\beta}{2}\|g\|.$$

Now, by the triangle inequality and the hypothesis, note that $\langle g, h + \delta \rangle \geq \alpha\|g\|\|h\| - \|g\|\|\delta\|$. Plugging in the above inequality, we get

$$\frac{\langle g, h + \delta \rangle}{\|h + \delta\|} \geq \alpha\|g\| \frac{\|h\|}{\frac{3\|h\|}{2}} - \|\delta\| \frac{\|g\|}{\frac{\beta}{2}\|g\|} \geq \frac{2\alpha\|g\|}{3} - \frac{2\|\delta\|}{\beta}. \qquad \square$$

## C   MISSING PROOFS IN SECTION 5

*Proof of Lemma 5.1.* We show that $\|P(x) - P(y)\| \leq \frac{\|x-y\|}{\alpha}$. The other claim follows symmetrically since $J^{-1}$ is the derivative of $P^{-1}$ at $P(x)$ and $x = P^{-1}(P(x))$.

Define $h(t) = x + t(y - x)$. Then we have:

$$\frac{d}{dt}\|P(h(t)) - P(h(0))\| = \left\langle \frac{P(h(t)) - P(h(0))}{\|P(h(t)) - P(h(0))\|}, J(h(t))h'(t) \right\rangle$$

$$\leq \|J(h(t))h'(t)\| \leq \frac{1}{\alpha}\|h'(t)\| = \frac{\|y - x\|}{\alpha},$$

where the second inequality uses the bound on the operator norm of $J$. Therefore, we have

$$\|P(y) - P(x)\| = \|P(h(1)) - P(h(0))\| \leq \int_{t=0}^{1} \frac{\|y - x\|}{\alpha} dt = \frac{\|y - x\|}{\alpha}. \qquad \square$$

*Proof of Lemma 5.2.* Let $\tilde{\mathcal{B}} = \{P(v) \mid v \in \mathcal{B}(x, \delta)\}$. Consider the line segment connecting $P(x)$ and $P(x_\star)$ and let $q$ be the intersection of this line segment with the boundary of $\tilde{\mathcal{B}}$. Then we must have $\|P^{-1}(q) - x\| = \delta$, and so by Lemma 5.1, $\|q - P(x)\| \geq \beta\delta$.

Next, $\|x\| \leq D$, $\|x_\star\| \leq D$, and $\|P^{-1}(q) - x\| = \delta$. By the triangle inequality we have $\|P^{-1}(q) - x_\star\| \leq \|P^{-1}(q) - x\| + \|x - x_\star\| \leq \delta + 2D \leq 3D$. By Lemma 5.1, we have $\|q - P(x_\star)\| \leq \frac{3D}{\alpha}$. Finally, since $q$ is on the line segment connecting $P(x)$ and $P(x_\star)$, it holds that

$$q - P(x_\star) = \frac{\|q - P(x_\star)\|}{\|P(x) - q\|}(P(x) - q).$$

Therefore, we have

$$
\begin{aligned}
f(x) - f(x_\star) &= C(P(x)) - C(P(x_\star)) \\
&= C(P(x)) - C(q) + C(q) - C(P(x_\star)) \\
&\leq C(P(x)) - C(q) + \langle \nabla C(q), q - P(x_\star) \rangle \\
&\leq C(P(x)) - C(q) + \frac{\|q - P(x_\star)\|}{\|P(x) - q\|} \langle \nabla C(q), P(x) - q \rangle \\
&\leq (C(P(x)) - C(q)) \left(1 + \frac{\|q - P(x_\star)\|}{\|P(x) - q\|}\right) \\
&\leq (C(P(x)) - C(q)) \left(1 + \frac{3D}{\alpha\beta\delta}\right) \\
&= (f(x) - f(P^{-1}(q))) \left(1 + \frac{3D}{\alpha\beta\delta}\right) \leq \epsilon \left(1 + \frac{3D}{\alpha\beta\delta}\right),
\end{aligned}
$$

where the first and third inequalities follow from the convexity of $C$ and the last inequality uses that $P^{-1}(q) \in \mathcal{B}(x, \delta)$. $\qquad\square$

*Proof of Theorem 5.4.* Notice that $I_1 \leq 4D_2\rho$. Since $\delta \leq 1/\rho$ and since $D_2 \geq \alpha\beta/\rho$, it holds that $I_k \leq \frac{40D_2}{\alpha\beta\delta_k}$ for all $k > 1$. Therefore, the total number of iterations is at most:

$$N = \sum_{k=1}^{K} I_k T_k = 8D_2\rho + \frac{40D_2\rho}{\alpha\beta} \sum_{k=2}^{K} 2^{3k/2} \leq 8D_2\rho + \frac{80D_2\rho}{\alpha\beta} 2^{3K/2}.$$

Next, by Lemma 5.3, we have:

$$\mathbb{E}[f(x_1^{K+1,1})] - f(x_\star) \leq 20\frac{HD_2}{\alpha\beta\sqrt{2^{K+1}}} \leq O\left(\frac{D_2^{4/3}H\rho^{1/3}}{(\alpha\beta)^{4/3}N^{1/3}}\right). \qquad\square$$

## C.1 Proof of Lemma 5.3

*Proof of Lemma 5.3.* We proceed by induction. For the base case, consider $k = 2$. From unrolling Lemma C.2, we have:

$$
\begin{aligned}
\mathbb{E}[f(x_1^{2,1})] - f(x_\star) &= \mathbb{E}[f(x_1^{1,T_1+1})] - f(x_\star) \\
&\leq \max\left\{\mathbb{E}[f(x_1^{1,1})] - f(x_\star) - I_1 \cdot \rho H\delta_1^2,\ 4\rho H\delta_1^2 \left(1 + \frac{4D_2}{\alpha\beta\delta_1}\right)\right\},
\end{aligned}
$$

since $f$ is hidden convex and $D_2 \geq R$, we have that $f(x_1^{1,1}) - f(x_\star) \leq HD_2$. Also $I_1 = 2D_2\rho = D_2/(\rho\delta_1^2)$, and so the first term above is at most 0. So, we have:

$$\leq 4\rho H\delta_1^2 \left(1 + \frac{4D_2}{\alpha\beta\delta_1}\right),$$

recalling that $D_2 \geq \alpha\beta/\rho$ and $\delta_1 \leq 1/\rho$ so that $1 \leq D_2/(\alpha\beta\delta_1)$:

$$\leq 20\frac{\rho HD_2\delta_1}{\alpha\beta} = 20\frac{HD_2}{\alpha\beta\sqrt{2}}.$$

Next, for the induction step, suppose $\mathbb{E}[f(x_1^{k,1}) - f(x_\star)] \leq 20\frac{HD_2}{\alpha\beta\sqrt{2^k}} = 20\frac{\rho HD_2\delta_k}{\alpha\beta}$ for some $k$. Then unrolling the result of Lemma C.2 yields:

$$\mathbb{E}[f(x_1^{k+1,1})] - f(x_\star) = \mathbb{E}[f(x_1^{k,T_k+1})] - f(x_\star)$$

$$\leq \max \left\{ \mathbb{E}[f(x_1^{k,1})] - f(x_\star) - I_k \cdot \rho H \delta_k^2, \ 4\rho H \delta_k^2 \left(1 + \frac{4D_2}{\alpha\beta\delta_k}\right) \right\}$$

$$= \max \left\{ \mathbb{E}[f(x_1^{k,1})] - f(x_\star) - I_k \cdot \rho H \delta_k^2, \ 4\rho H \delta_k^2 \left(1 + \frac{4D_2}{\alpha\beta\delta_k}\right) \right\}$$

using $D_2 \geq \alpha\beta/\rho$ and $\delta_k \leq 1/\rho$:

$$\leq \max \left\{ \mathbb{E}[f(x_1^{k,1})] - f(x_\star) - I_k \cdot \rho H \delta_k^2, \ 20\frac{\rho H \delta_k D_2}{\alpha\beta} \right\}$$

$$\leq \max \left\{ \mathbb{E}[f(x_1^{k,1})] - f(x_\star) - I_k \cdot \rho H \delta_k^2, \ 20\frac{H D_2}{\alpha\beta\sqrt{2^k}} \right\}$$

notice that $I_k \geq \frac{20D_2}{\alpha\beta\delta_k}$ and use the induction hypothesis:

$$\leq 20\frac{H D_2}{\alpha\beta\sqrt{2^k}}.$$

This completes the induction. $\qquad\square$

**Lemma C.1.** *Let $a$ and $b$ be arbitrary points in $\mathbb{R}^d$. Then, we have*

$$\|P(a) - P(b) - J(b)(a - b)\| \leq \frac{\rho\|b - a\|^2}{2}.$$

*Proof.* Let $v$ denote an arbitrary unit vector. We define the function $s : \mathbb{R} \to \mathbb{R}$ by $s(\lambda) = \langle v, P(a\lambda + b(1 - \lambda)) \rangle$. Then $s'(\lambda) = \langle v, J(a\lambda + b(1 - \lambda))(a - b) \rangle$. Therefore:

$$\langle v, P(a) - P(b) \rangle = s(1) - s(0) = \int_0^1 s'(\lambda)\, d\lambda$$

$$= s'(0) + \int_0^1 (s'(\lambda) - s'(0))\, d\lambda$$

$$= \langle v, J(b)(a - b) \rangle + \int_0^1 (s'(\lambda) - s'(0))\, d\lambda.$$

$$|\langle v, P(a) - P(b) - J(b)(a - b) \rangle| \leq \int_0^1 |\langle v, (J(a\lambda + b(1 - \lambda)) - J(b))(a - b) \rangle|\, d\lambda$$

$$\leq \rho\|a - b\|^2 \int_0^1 \lambda\, d\lambda = \frac{\rho\|a - b\|^2}{2}.$$

Since this holds for all $v$, we have that $\|P(a) - P(b) - J(b)(a - b)\| = \sup_{\|v\|=1} |\langle v, P(a) - P(b) - J(b)(a - b) \rangle| \leq \frac{\rho\|b - a\|^2}{2}$. $\qquad\square$

**Lemma C.2.** *Algorithm 3 ensures:*

$$\mathbb{E}[f(x_1^{k,i+1})] - f(x_\star) \leq \max \left\{ \mathbb{E}[f(x_1^{k,i})] - f(x_\star) - \rho H \delta_k^2, \ 4\rho H \delta_k^2 \left(1 + \frac{4D_2}{\alpha\beta\delta_k}\right) \right\}.$$

*Proof.* To simplify notation, let $x_t = x_t^{k,i}$, $\overline{x} = \overline{x^{k,i}}$, $\mathcal{X} = \mathcal{X}^{k,i}$, $T = T_k$, and $\delta = \delta_k$. We also define $J_t = J(x_t)$ and $g_t = \nabla C(y_t)$ where $y_t = P(x_t)$. Then we have $\mathbb{E}[h_t] = J_t^\top g_t$.

Notice that by assumption on $D_1$ and $D_2$, $f(x_1^{k,i+1}) \leq f(\overline{x})$. So, it suffices to show that:

$$\mathbb{E}[f(\overline{x})] - f(x_\star) \leq \max \left\{ \mathbb{E}[f(x_1)] - f(x_\star) - \rho H \delta^2, \ 4\rho H \delta^2 \left(1 + \frac{4D_2}{\alpha\beta\delta}\right) \right\}.$$

Then, for all $u \in \mathcal{X}$ we have:

$$\sum_{t=1}^T f(x_t) - f(u) = \sum_{t=1}^T C(P(x_t)) - C(P(u))$$

$$\leq \sum_{t=1}^{T} \langle \nabla C(P(x_t)), P(x_t) - P(u) \rangle$$

$$= \sum_{t=1}^{T} \langle \nabla C(P(x_t)), P(x_t) - P(u) - J_t(x_t - u) \rangle + \sum_{t=1}^{T} \langle \nabla C(P(x_t)), J_t(x_t - u) \rangle.$$

By Lemma C.1 and recalling $\|\nabla C(P(x_t))\| \leq H$ and $\|x_t - u\| \leq 2\delta$:

$$\leq 2\rho T H \delta^2 + \sum_{t=1}^{T} \langle \nabla C(P(x_t)), J_t(x_t - u) \rangle$$

$$= 2\rho T H \delta^2 + \sum_{t=1}^{T} \langle J_t^\top \nabla C(P(x_t)), x_t - u \rangle$$

$$= 2\rho T H \delta^2 + \sum_{t=1}^{T} \langle h_t, x_t - u \rangle + \sum_{t=1}^{T} \langle \mathbb{E}[h_t] - h_t, x_t - u \rangle.$$

Note that the points $\{x_t\}$ are chosen via projected online gradient descent where the domain has radius $\delta$. Hence the linearized regret is upper bounded by $\sum_{t=1}^{T} \langle h_t, x_t - u \rangle \leq \delta H \sqrt{T}$ (see, e.g., Zinkevich, 2003).

$$= 2\rho T H \delta^2 + \delta H \sqrt{T} + \sum_{t=1}^{T} \langle \mathbb{E}[h_t] - h_t, x_t - u \rangle,$$

$$\mathbb{E}\left[\sum_{t=1}^{T} f(x_t) - f(u)\right] \leq 2\rho T H \delta^2 + \delta H \sqrt{T} = 3\rho H \delta^2 T,$$

where the last equality uses $\delta = \frac{1}{\rho\sqrt{T}}$. Next, notice that by definition,

$$\mathbb{E}[f(\bar{x})] = \mathbb{E}\left[\frac{1}{T}\sum_{t=1}^{T} f(x_t)\right].$$

Therefore, we have that for all $u \in \mathcal{X}$ :

$$\mathbb{E}[f(\bar{x}) - f(u)] \leq 3\rho H \delta^2.$$

Now, consider two cases.

*Case 1:* $f(x_1) - f(u) \geq 4\rho H \delta^2$ for some $u \in \mathcal{X}$. Here, we have:

$$\mathbb{E}[f(\bar{x}) - f(x_1)] = \mathbb{E}[f(\bar{x}) - f(u) + f(u) - f(x_1)] \leq 3\rho H \delta^2 - 4\rho H \delta^2 = -\rho H \delta^2,$$

$$\mathbb{E}[f(\bar{x}) - f(x_\star)] = \mathbb{E}[f(x_1) - f(x_\star)] + \mathbb{E}[f(\bar{x}) - f(x_1)] \leq \mathbb{E}[f(x_1)] - f(x_\star) - \rho H \delta^2.$$

*Case 2:* $f(x_1) - f(u) < 4\rho H \delta^2$ for all $u \in \mathcal{X}$. Then since we have $\|x_1\| \leq D_2$, $\|x_\star\| \leq D_2$, by Lemma 5.2, we have:

$$f(\bar{x}) - f(x_\star) = f(\bar{x}) - f(x_1) + f(x_1) - f(x_\star) \leq 4\rho H \delta^2 \left(1 + \frac{4D_2}{\alpha\beta\delta}\right).$$

Thus, in both cases, we have:

$$\mathbb{E}[f(\bar{x})] - f(x_\star) \leq \max\left\{\mathbb{E}[f(x_1)] - f(x_\star) - \rho H \delta^2, \; 4\rho H \delta^2 \left(1 + \frac{4D_2}{\alpha\beta\delta}\right)\right\}. \qquad \square$$

