# OpenReview forum: "Descent with Misaligned Gradients and Applications to Hidden Convexity"
_ICLR.cc/2025/Conference — ICLR 2025 Poster_

### Official Review · Reviewer_HMLj · 2024-10-27

**Soundness:** 3
**Presentation:** 3
**Contribution:** 3
**Rating:** 8
**Confidence:** 4

**Summary:**

The paper considers the problem of first-order stochastic optimization where the learner can only access a stochastic gradient whose expectation is only guaranteed to be correlated with (but not equal to) the true gradient at the queried point.  The authors consider three different settings commonly encountered in machine learning problems where the learner can only access biased gradients. For each of the three settings, they propose a new algorithm and provide its analysis.

**Strengths:**

I like the paper overall. It uses simple but interesting additions to existing strategies for optimization to design the algorithm with biased gradient estimates which often yield optimal performance. I think the results are sufficiently novel and interesting and improve upon the best known results so far.

**Weaknesses:**

I don't see any glaring weakness in the paper but I have some questions listed in the next section.

**Questions:**

1. Can the authors explain the intuition behind the update in line 6 in algorithm 2? It seems like you just want to consider the update in the orthogonal direction but I can't quite understand why? Is it simply to reduce the norm of the update (and not considering the update along $x\_t$ helps) or is there something fundamental that is going on there?

2. Does strong-convexity help for algorithm 2 and 3? In other words, if we are given the additional information of strong convexity, how much does that help improve the error? Particularly for alg 2, if it does not help then what exactly from the analysis in Demidovich et al, 2023 does not work out in this case?

3. I am not sure if this will work, but might be worth a try: If I understand correctly, the term $f(x_t) - f(x^{\star})$ in line 413 cancels out with the term $-\frac{\eta\_t \alpha \| g\_t \| }{3}$ in line 416. I think with an appropriate change of constants, we can retain a fraction of the negative term in line 416 and carry forward it to the equation in line 421. Now, if $\|g\_t\| \leq \frac{C}{\sqrt{t}}$ for some $C > 0$, then we are at a point with small gradient. Otherwise, it will cancel out the $\frac{1}{\sqrt{B_t}}$ term in the equation in line 421 with $B_t = t + 1 + k$. This might help you achieve optimal error rate. Of course this needs to be checked but it might be helpful to address the sub-optimality gap.

---

> ### Author Response · Authors · 2024-11-19
> **reply**
>
> Thank you for your positive review! Below we address your questions:
>
> 1. The update in line 6 is essentially performing a projection. It’s possible that replacing this line with an ordinary projection to the sphere of radius $D$ would yield a similar final result, but it is not clear how to show such a thing. Instead, line 6 adds a correction to the update that still guarantees that the iterates do not grow too large (similar to projections), but can be integrated more easily into the calculations.
> 2. Yes, strong convexity should help in these cases. However, notice that unlike in standard convex optimization settings, we cannot simply regularize a non-strongly convex objective to become strongly convex because this would eliminate the correlation property and it is not clear what effect this would have on the analysis.
> 3. This is a nice idea and thank you for the suggestion, but unfortunately, it does not seem to help. Notice that the *total number of gradient evaluations* is $N=\sum_{t=1}^T B_t$, so with $B_t=\Theta(t)$, we’d have $N=\Theta(T^2)$. Thus, a convergence bound of $O(1/\sqrt{T})$ is actually $O(1/N^{1/4})$, which is worse than our $O(1/N^{1/3})$ result.  Please let us know if we are missing something here!

---

> > ### Comment · Reviewer_HMLj · 2024-11-22
> > **Response to authors**
> >
> > 1. I understand it is a projection but I don't understand an intuitive reason for that particular choice of update. It does not seem close enough to just a projection on the sphere.
> >
> > 2. I agree, we cannot add a strongly convex regularization or something similar. However, if it were known that the underlying function is strongly convex to begin with, then what is the benefit in terms of convergence guarantees?
> >
> > 3. I see what happens. I think I mistook $T$ for $N$. Another thing that might work is to use an epoch based approach where the batch size is kept constant throughout the epoch and doubled every epoch. The advantage in doubling is that the number of changes becomes $\log T$ as opposed to $T$, which might prove to be helpful. In my experience with similar analyses where I  have run into the same issue, it seems like exponential doubling can often be quite helpful.

---

> > > ### Author Response · Authors · 2024-12-02
> > > **reply**
> > >
> > > 1. Taking a look at Lemma 4.1, the idea is that we first eliminate the component of $h_t$ that is parallel to $x_t$ if that component is pushing the update in the “wrong direction” (i.e., away from the origin). This will keep $x_t$ from getting too much bigger, but still even a purely perpendicular update results in a small increase in norm. So, we add a small $-\eta^2 x_t/\|x_t\|^2$ update to counter this effect. This gives us a concrete formula for the update that keeps the norm from growing, but also because the update has a somewhat simple formula, we can argue is still essentially correlated with the true gradient and so allows for easier later analysis.  Thanks for asking - we will add this explanation in the revision.
> > >
> > > 2. It’s actually easier to go from analysis for smooth losses (such as ours) to analysis for strongly-convex losses in general. The classical approach uses the fact that the distance-to-optimality-squared is less than the function gap. This shows that if $\|x_1-x_\star\|\le 2^{-k}$,  then after $N=O(2^{3k}/\mu^{3})$ gradient evaluations, we can ensure $\mathbb{E}[\|x_T-x_\star\|]\le 2^{-k+1}$. Repeating this for $K$ epochs costs $M=2^{3K}/\mu^{3}$ iterations and yields a distance of $O(2^{-K})$,  which implies a suboptimality  gap of $O(2^{-2K})=O(1/(\mu M)^{2/3})$ (since the loss is smooth). This is not as good as the $O(1/\mu M)$ possible with standard unbiased gradient estimates, but satisfies the intuition that strong-convexity is helpful for optimization. It is incomparable with the $O(1/\mu^2 M)$ achieved by Demidovich et al 2023: the dependency on $M$ is worse, but the dependency on $\mu$ is better. Thanks for suggesting this discussion!
> > >
> > > 3. Epochs and doubling: thanks for the suggestion; we did try similar approaches. For example, our current bounds can also be obtained by considering \log T epochs, where in epoch $j$, the learning rate is ~ 1/2^j and the batch size is 2^{2j}. This is intuitively also why we have a \log T in our bound. While it is, of course, possible that we missed some parameter setting, we believe improving over T^{1/3} will require a new idea, either a new potential or something algorithmically novel.

---

### Official Review · Reviewer_7H59 · 2024-11-03

**Soundness:** 3
**Presentation:** 3
**Contribution:** 3
**Rating:** 6
**Confidence:** 3

**Summary:**

The paper studies stochastic convex optimization where the stochastic gradient oracle is biased but correlated with the true gradient. The proposed algorithms achieve the following performances: for Lipschitz, convex objectives and slowly varying bias, the rate is O(N^{-1/2}); for Lipschitz, smooth convex objectives and general correlated stochastic gradient oracle, the rate is O(N^{-1/3}). The results are applied to problems with hidden convexity, which achieves a rate O(N^{-1/3}).

**Strengths:**

The paper is overall well written, with clearly presented setups, algorithms, and performance. In addition, the correlated stochastic oracle studied, as pointed out on page 2, might have broad applications.

**Weaknesses:**

-- The paper is closely related to the stochastic optimization literature. Although the authors have cited many relevant works, the exact known results are missing in this paper. It might help the readers better appreciate the significance of the results by providing more details and/or comparisons with existing setups and known upper/lower bounds on the convergence rate.

-- The assumptions of each theorem are stated at the beginning of each corresponding section (informally). It might be better to present them more formally, either as Asssumption 1/2/3, or stated directly in the theorems.

-- In terms of significance, it is unclear how tight the bounds are. Would it be possible to derive some lower bounds from known results for other related problems? This would greatly help the readers appreciate the significance of the results. In addition, it seems that the analysis is relatively standard. Could the authors provide more comparisons with existing proofs for stochastic convex optimization, or related problems/setups?

**Questions:**

-- All the convergence results are presented in expectation. I’m wondering how hard it is to obtain ``with high probability’’ performance guarantee?

-- In line 113, ``note that this is equivalent to the condition that …”, this seems to require that f is differentiable. Otherwise, the gradient of f may not exist, and instead the bound holds for all subgradients.

---

> ### Author Response · Authors · 2024-11-19
> **reply**
>
> Thanks very much for your review. Below we address some of the points you have raised:
>
> Regarding comparison to known bounds: In the setting of Section 3, we are not aware of any works that  consider exactly our setting, but we note that the dependence on $N$ is certainly tight due to standard bounds with unbiased gradients. As discussed in the response to reviewer SpNX, our results in Section 3 and 4 improve upon those of Ajalloeian & Stich 2020 by either weakening assumptions of improving the convergence bound from $O(1/N^{1/4})$ to $O(1/N^{1/3})$. In Section 5, we improve upon the $O(1/N^{1/4})$ rate obtained by Chen et al. 2022 to $O(1/N^{1/3})$.
>
> Regarding the assumption statements: Thank you for your suggestion.  In the revision we will present them formally as numbered Assumptions and refer to them in the Theorem restatements.
>
> Regarding tightness: our first result is tight in the dependence on $N$ ($O(1/N^{1/2})$) as this is the lower bound even for exact gradient oracles. For the other results (or the dependencies on the $\mu$ and $\lambda$ parameters), our results are the best in their respective classes, but we are not aware of any lower bounds that apply.
>
> Regarding the analysis novelty: Our analysis is unique in several respects. For example, our analysis for Section 3.2 makes critical use of a “stability” property enjoyed by anytime averaging (but not by the “standard” Polyak averaging). This stability is very rarely present in the literature. We also have never seen any result similar to our technique for viewing projection onto a sphere with the $\ell_2$-norm as projection onto a  different unknown convex set using an unknown norm of interest (Lemma 3.1). In Sections 4 and 5, we introduce novel techniques to control the norm of the updates that were more analytically tractable than standard projections.
>
> Regarding the questions:
> 1. Obtaining high-probability statements should be a fairly straightforward exercise, but it would unduly complicate the analysis.  Note that all of our results involve summing up expected values of certain quantities. The difference between the realized value and the expected value is thus  controllable with standard martingale concentration (e.g., Azuma–Hoeffding) bounds. Thank you pointing this out - we will remark on this in the revision.
> 2. Yes, you are correct. In our analysis $\nabla f$ can indicate an arbitrary subgradient whenever the gradient does not exist.  Thank you for noticing this.  We will clarify this point in the revision.

---

> > ### Comment · Reviewer_7H59 · 2024-11-25
> >
> > Thank the authors for the comments! I'll keep my score.

---

### Official Review · Reviewer_hxRG · 2024-11-04

**Soundness:** 3
**Presentation:** 2
**Contribution:** 2
**Rating:** 6
**Confidence:** 3

**Summary:**

The paper studies oracle-based optimization in three settings. First, when the oracle returns gradients that are misaligned with the true gradients in a specific manner: the expectation of the returned gradient is positively correlated with the true gradient (in terms of the inner product). Second, for more specific applications, they strengthen this assumption, and require that the lower bound not just be nonzero, but that it is at least the square of the true gradient. Third, for their setting of hidden convexity, they use the standard unbiased estimator assumption.

The paper provides improved rates of convergence under all three settings.

**Strengths:**

I think it's a well-motivated paper with a coherent set of results. I like that the analysis in Section 3, though, simple, is complete and step-by-step. The authors are also quite honest about differences with prior work, though they could do a better job explaining why they do better than existing work in similar assumptions (see Questions).

**Weaknesses:**

My only complaint is that the paper's introduction suggests that the only assumption made is that the inner product of the true gradient with the expected gradient provided by the oracle is positive: however, this seems to hold only in Section 3. In Section 4, this is strengthened to say that the lower bound is the squared norm of the true gradient (an assumption same as Beznosikov et al), and in Section 5, it's further strengthened to be simply an unbiased estimator. Is my understanding accurate? If so, the "misaligned" description used throughout the introduction applies only to Section 3, and for the other results, there already exists standard terminology for those assumptions, so assigning them a new name wouldn't be the right thing to do.

My recommendation to the authors is to please clarify all the assumptions (for each of the different settings studied) in the introduction, so as to avoid any confusion.

Further, it would be useful to have a better understanding of what specific difference in the analyses in Section 4 lead to the improved rates as compared to existing work under this assumption (see Questions).

**Questions:**

1. Does the analysis in Section 3 have any connection with the analysis seen when using self-concordant barriers? The assumption that two matrices $A_t$ and $A_{t+1}$ do not change much is quite similar to saying that two successive Hessians do not (which is essentially what self-concordance captures). If the authors believe there could be a connection to this, it would be useful to add that to the paper and add pointers to the literature on interior-point methods, where this notion is used; if not, then it would still help to clarify why it differs.

2. The assumption in Section 4 that the inner product of the true gradient and expected gradient (from the oracle) is lower bounded by the square of the true gradient norm is identical to that in Beznosikov et al (as the authors themselves note). Can the authors explain what exactly they do differently to improve the $\epsilon^{-4}$ rate to $\epsilon^{-3}$? Could they point to a specific step in their proof where they use this inner product assumption in a better manner?

3. There was a recent paper https://arxiv.org/abs/2304.08596 by Shu, Ramachandran, and Wang, which also talks about hidden convexity. I think it would be useful to cite the paper if the way the phrase "hidden convexity" is used is the same. If not, it would be helpful to clarify the differences.

---

> ### Author Response · Authors · 2024-11-19
> **reply**
>
> Thank you very much for your comments; they will be very helpful in improving our work!
>
> Regarding the use of misaligned gradients, we believe that there is a misunderstanding. In all three sections of the paper, we consider misaligned gradients. Specifically, in
> * Section 3, we assume that the expected gradient is obtained by multiplying the true gradient with a PSD matrix (unobserved); further, these matrices do not change much over time.
> * Section 4, we consider a more general setup, where the expected gradient is simply assumed to be correlated with the true gradient. We also require a lower bound on the norm of the expected gradient.
> * Section 5, we consider the setting of hidden convexity – here we wish to minimize a (non-convex) function f that can be expressed as a convex function after a non-linear transformation, i.e., $ f(x) = C(P(x))$ where $C$ is convex and $P$ is a non-linear coordinate transform. Here, we obtain an unbiased estimate of the gradient of $f$; however, due to the transformation, this can be viewed as a misaligned gradient for $C$.
>
> In the revision we will clarify our setting to avoid this confusion.
>
> Regarding the questions,
> 1. It’s possible that there is a connection here, but it is certainly not obvious as there are several key differences in our usage. For example, we do not use the matrices as explicit preconditioners to form updates because we never actually observe these matrices at all!
> 2. Note that Beznosikov et al. assume strongly-convex losses, which we do not assume. As discussed in our paper, the standard conversion from non-strongly convex to strongly-convex via regularization does not obviously apply here. Moreover, it appears that the assumptions of Beznosikov et al. are actually stronger than ours - their Definition 2 in fact forces gradient estimates that decay to zero when the true gradient goes to zero. Our setting explicitly does *not* require this and so is able to  obtain asymptotic convergence even in the presence of persistent bias. Thank you for bringing this to our attention - we will incorporate this discussion into the revision.
> 3. The use of hidden convexity by Shu, Ramachandran, and Wang is very different from ours. They consider a convex objective with a constraint that is non-convex, but for which it is possible to replace the constraint with a convex constraint without changing the objective value. Thus, the problem can be reformulated as a convex optimization problem. Our problem *cannot* be reformulated as a convex optimization problem.

---

> > ### Comment · Reviewer_hxRG · 2024-11-21
> > **Thanks!**
> >
> > Thanks to the authors for their clarifications. I'll keep my score.

---

### Official Review · Reviewer_SpNX · 2024-11-08

**Soundness:** 3
**Presentation:** 3
**Contribution:** 3
**Rating:** 6
**Confidence:** 2

**Summary:**

This paper focuses on the case where the stochastic oracle produces feedback which is not necessarily unbiased. More precisely, it introduces the notion of misaligned stochastic gradients in order to capture the lack unbiasedness in several practical scenarios. To that end, the authors test their theoretical machinery for the optimization problems with hidden convexity (also studied in Sakos 2024 and references therein) and provide an algorithmic method which exhibit $\mathcal{O}(\varepsilon^{-3})$ iteration complexity.

**Strengths:**

The paper is very well written and easy to follow. Moreover, the mathematical analysis is sound and clear as far I have checked. The idea of misaligned stochastic vectors is quite intuitive and as far as my knowledge goes it paves the way for dealing with a useful practical methodology for structured biased stochastic gradients.

**Weaknesses:**

Concerning this paper I have two main concerns/questions:

1. The almost sure boundedness of the biased gradients seems to be a quite restrictive statistical assumption. As far as my knowledge this type of assumption is usually used in methods which are run with adagrad-type step-sizes (see for example Levy 2017). Thus, my question is two-fold: Does this statistical assumption hold in practice and secondly do the authors believe that it is an artefact of the analysis or the method in order to overcome it ?

2. The paper lacks a numerical comparison with other methods which consider biased gradients like the Stich 2020 paper. My question concerns the fact that the compression scheme presented in the said paper seems to cover the case of an "relative bias" (an analogy to Polyak's relative noise) in the sense that the bias vanishes when we approach a solution. To that end, some simple calculations may show that under this condition the second assumption in oracle & assumptions may be recovered. So, I think that a more thorough discussion is needed.

**Questions:**

See weaknesses.

---

> ### Author Response · Authors · 2024-11-19
> **reply**
>
> Thank you for your work reviewing our paper! Below we answer your questions:
>
> 1. Thank you for pointing this out.  Fortunately, the boundedness of the biased gradients holds in practice on many neural network tasks.  See, e.g., the recent work of [Defazio et al.,](https://arxiv.org/pdf/2310.07831) where in Figure 3 they show that the assumption holds for commonly-studied models such as wide ResNet, IWSLT14, GPT, RoBERTa, DLRM, MRI, ViT, RCNN. (See also Figure 3 in [Zhao et al., ICML 2022](https://proceedings.mlr.press/v162/zhao22i/zhao22i.pdf) and Figure 3 in [Xiong et al., ICML 2020](https://arxiv.org/pdf/2002.04745).) Note that even with this assumption, our analysis turns out to be rather nontrivial. Removing the boundedness assumption from our algorithm/analysis is an interesting open question and we will mention it in the revision.
>
> 2. The setting of Ajalloeian & Stich 2020 still requires that the bias is “smaller” than the true gradient (their $m\le 1$ condition in Assumption 4).  While our setting 3 also requires the bias to vanish as the gradient approaches zero, it does *not* require the bias to be smaller than the gradient. This is a significant difference that makes analysis much more complicated because intuitively we cannot rely on the “sign” of the biased gradient to tell us the sign of the true gradient. Moreover, their bounds are worse: the relevant comparison is their Theorem 4, which implies convergence in gradient norm squared at a rate of $O(1/\epsilon^2)$, which implies function value convergence at a rate $O(1/\epsilon^4)$.
>
> In contrast, in Section 4, our setting explicitly allows the bias to *not* vanish when we approach a solution, so it is unlikely that the condition in Ajalloeian & Stich 2020 would imply our assumptions. We consider this to be an interesting feature of the misalignment assumption: it provides a natural way to model non-vanishing bias that nevertheless allows for asymptotic convergence guarantees.
>
> Thank you for suggesting these detailed comparisons - we will add them to the discussion in the paper!

---

> > ### Comment · Reviewer_SpNX · 2024-12-02
> > **Thank you for the reply**
> >
> > I thank the authors for their responses. My concerns are clarified therefore I am willing to keep my score.

---

### Meta-Review · Area_Chair_MGer · 2024-12-09

**Metareview:**

This paper introduces a novel optimization approach for convex objectives using misaligned stochastic gradients, where the gradients are correlated but not equal to the true gradient. The authors propose algorithms with iteration complexities of $\widetilde{O}(\epsilon^{−2})$ and $\widetilde{O}(\epsilon^{−3})$, depending on the rate of misalignment, and apply the framework to hidden convexity problems.

Overall, the paper provides valuable theoretical contributions, particularly the introduction of the misaligned gradient framework, which has potential practical applications. The clarity of the mathematical analysis and the novel handling of biased gradients make this paper a valuable addition to optimization literature.

**Additional Comments On Reviewer Discussion:**

There are some concerns regarding the clarity of the assumptions, especially in relation to the term "misaligned". The lack of empirical validation and direct comparison with existing methods (e.g., Stich 2020) also limits the paper’s practical applicability. The authors have responded to these concerns with clarifications and further comparisons.

---

### Decision · Program_Chairs · 2025-01-22

Accept (Poster)